# Adverse Crosstalk between Extracellular Matrix Remodeling and Ferroptosis in Basal Breast Cancer

**DOI:** 10.3390/cells12172176

**Published:** 2023-08-30

**Authors:** Christophe Desterke, Emma Cosialls, Yao Xiang, Rima Elhage, Clémence Duruel, Yunhua Chang, Ahmed Hamaï

**Affiliations:** 1UFR Médecine-INSERM UMRS1310, Université Paris-Saclay, F-94800 Villejuif, France; 2Institut Necker Enfants Malades, INSERM UMR-S1151-CNRS UMR-S8253, Université Paris Cité, F-75015 Paris, France; emma.cosialls@inserm.fr (E.C.); xiangyao.ecole@outlook.com (Y.X.); rima.elhage@inserm.fr (R.E.); clemence.duruel@inserm.fr (C.D.); yunhua.chang-marchand@inserm.fr (Y.C.); 3Team 5/Ferostem Group, F-75015 Paris, France

**Keywords:** basal breast cancer, extracellular matrix remodeling, ferroptosis, transcriptome, text mining

## Abstract

(1) Background: Breast cancer is a frequent heterogeneous disorder diagnosed in women and causes a high number of mortality among this population due to rapid metastasis and disease recurrence. Ferroptosis can inhibit breast cancer cell growth, improve the sensitivity of chemotherapy and radiotherapy, and inhibit distant metastases, potentially impacting the tumor microenvironment. (2) Methods: Through data mining, the ferroptosis/extracellular matrix remodeling literature text-mining results were integrated into the breast cancer transcriptome cohort, taking into account patients with distant relapse-free survival (DRFS) under adjuvant therapy (anthracyclin + taxanes) with validation in an independent METABRIC cohort, along with the MDA-MB-231 and HCC338 transcriptome functional experiments with ferroptosis activations (GSE173905). (3) Results: Ferroptosis/extracellular matrix remodeling text-mining identified 910 associated genes. Univariate Cox analyses focused on breast cancer (GSE25066) selected 252 individual significant genes, of which 170 were found to have an adverse expression. Functional enrichment of these 170 adverse genes predicted basal breast cancer signatures. Through text-mining, some ferroptosis-significant adverse-selected genes shared citations in the domain of ECM remodeling, such as TNF, IL6, SET, CDKN2A, EGFR, HMGB1, KRAS, MET, LCN2, HIF1A, and TLR4. A molecular score based on the expression of the eleven genes was found predictive of the worst prognosis breast cancer at the univariate level: basal subtype, short DRFS, high-grade values 3 and 4, and estrogen and progesterone receptor negative and nodal stages 2 and 3. This eleven-gene signature was validated as regulated by ferroptosis inductors (erastin and RSL3) in the triple-negative breast cancer cellular model MDA-MB-231. (4) Conclusions: The crosstalk between ECM remodeling-ferroptosis functionalities allowed for defining a molecular score, which has been characterized as an independent adverse parameter in the prognosis of breast cancer patients. The gene signature of this molecular score has been validated to be regulated by erastin/RSL3 ferroptosis activators. This molecular score could be promising to evaluate the ECM-related impact of ferroptosis target therapies in breast cancer.

## 1. Introduction

In 2020, breast cancer was the most common cancer diagnosed in women in the United States [1]. Breast cancer is a heterogeneous disease with different molecular subtypes, defined by distinct molecular classes associated with the prognosis: claudin-low, normal-like, luminal A, luminal B, HER2, and basal [2], and confirmed via gene quantification in pam50 classification [3]. Breast cancer is the second leading cause of mortality in women due to rapid metastasis and disease recurrence [4]. Breast tissue is in a unique microenvironment, with plentiful adipocytes infiltrating. Previous studies have shown that adipocytes can regulate fatty acid metabolism, and enhance the invasion and metastasis of breast cancer [5]. Ferroptosis is an iron-dependent regulated form of cell death caused by the accumulation of lipid-based reactive oxygen species (ROS) [6]. Prerequisites for ferroptosis include iron metabolism, mitochondrial metabolism, synthesis of polyunsaturated fatty acid phospholipid (PUFA-PL), and lipid peroxidation [7]. Therefore, targeting ferroptosis has been proposed to treat breast cancer. There is increasing evidence that ferroptosis can inhibit breast cancer cell growth, improve the sensitivity of chemotherapy and radiotherapy, and inhibit distant metastases [5]. From 2001 to 2003, the Stockwell Lab performed a screen to identify compounds that kill cells engineered to be tumorigenic (harboring the RAS mutant) without killing their isogenic parental precursors. One of the most efficient compounds was identified and named “erastin” due to its ability to “Eradicate RAS-and Small T transformed cells” [8]. Subsequently, they identified RSL3, which was also named after its “oncogenic RAS-selective lethal” property in 2008 [9]. Small molecule-induced ferroptosis has been shown to have a strong inhibitory effect on tumor growth in a drug-resistant environment, which may increase the sensitivity of the tumor to chemotherapeutic treatment [10]. Ferroptosis is also considered an important cell death mechanism caused by several therapies, including chemotherapy, radiotherapy (RT), targeted therapy, and immunotherapy [11]. Still, contrarily, the tumor cells with ferroptosis could diminish anti-tumor immune response by inhibiting the antigen-presenting cells [12]. The tumor microenvironment (TME) plays a notable role in cancer progression. It includes pH and oxygen levels, the extracellular matrix (ECM), connective tissue, infiltrating immune cells, and the vasculature of the tumor. Interaction between the ECM and the tumor cells activates key signaling pathways that promote tumor proliferation, invasion, and metastasis. This notably influences many tumors, as the ECM can comprise up to 60% of the tumor mass [13].

In the present work, through a text-mining approach integrated into transcriptome experiments, a link between ferroptosis and ECM remodeling has been established through gene-related regulation in the adverse prognosis of breast cancer but also in TNBC cellular model stimulated by ferroptosis activators.

## 2. Materials and Methods

### 2.1. Determination of Ferroptosis in Breast Cancer-Related Genes

Using the keywords “Ferroptosis in breast”, a co-occurrence of citations with coding gene identifiers was searched for in the article abstracts of the PUBMED database with the “Génie” algorithm [14]. Bioinformatics analyses were realized in R software environment version 4.2.1. Further investigations of text-mining associations with gene identifiers have been confirmed with the “GeneValorization” application [15] to the National Center for Biotechnology Information (NCBI) database [16]. The results of this text-mining were drawn as circosplot of gene-keywords co-occurrence with circlize R-package version 0.4.15 and as alluvial plot.

### 2.2. Transcriptome Cohort of Breast Cancer for Patients Treated with Anthracyclin and Taxanes

Transcriptome normalized matrix of dataset GSE25066 [17] was downloaded at the following address: https://www.ncbi.nlm.nih.gov/geo/query/acc.cgi?acc=gse25066 (accessed on 18 May 2023) and annotated with the corresponding technology platform GPL96 (HG-U133A) Affymetrix Human Genome U133A Array available at the following address: https://www.ncbi.nlm.nih.gov/geo/query/acc.cgi?acc=GPL96 (accessed on 18 May 2023).

### 2.3. Multi-Omics Validation Breast Cancer Cohort METABRIC

A multi-omics validation cohort of METABRIC [18,19,20] comprising 1666 samples of breast cancer samples was analyzed through Cbioportal web server [21]. This application allowed for validating at multi-omics level (transcription, mutation, methylation) the eleven-gene signature selected in the training cohort according relapse-free survival outcome, but also the associated breast cancer subtype clinical parameters associated.

### 2.4. Transcriptome Dataset Testing Effect of Ferroptosis Inducers on Triple-Negative Breast Cancer Cell Models

Fragments per kilobase of transcript sequence per million base pairs sequenced (FPKM) transcript quantification performed using the original pipeline of dataset GSE173905 [22] was downloaded at the following address: https://www.ncbi.nlm.nih.gov/geo/query/acc.cgi?acc=GSE173905 (accessed on 18 May 2023). After sequencing on Illumina NovaSeq 6000 technology, the original pipeline-aligned reads on the human genome with the reference genome were built using Hisat2 v2.0.5 [23], and paired-end clean reads were aligned to the reference genome using the Hisat2 v2.0.5 software and FPKM, expected number of fragments per kilobase of transcript sequence per million base pairs sequenced were computed on the counts obtained with Feature Counts v1.5.0-p3 software [24]. 

The matrix of RNASEQ FPKM quantification from the dataset GSE162069 [25] was downloaded at the following web address: https://www.ncbi.nlm.nih.gov/geo/query/acc.cgi?acc=GSE162069 (accessed on 23 June 2023). Library preparation was performed using the Truseq stranded mRNA library kit (Illumina, Illumina Inc., San Diego, CA, USA), followed by poly-T-based RNA purification beads. Sequencing was performed on HiSeq2500 (Illumina, Illumina Inc., San Diego, CA, USA). The reads were aligned with the Tophat algorithm [26] and transcript quantification was carried out with Cufflinks [27].

The normalized microarray matrix from dataset GSE154425 [28] was downloaded at the following address: https://www.ncbi.nlm.nih.gov/geo/query/acc.cgi?acc=GSE154425 (accessed on 23 June 2023). The normalized data were gene annotated with technology platform GPL17692: Affymetrix Human Gene 2.1 ST Array (Affymerix, Santa Clara, CA, USA) available at the following address: https://www.ncbi.nlm.nih.gov/geo/query/acc.cgi?acc=GPL17692 (accessed on 23 June 2023). 

On the selected genes for Ferroptosis/Extracellular matrix remodeling signature, unsupervised principal component analysis was performed with FactoMiner R package version 2.8 [29] on the three respective datasets: GSE154425, GSE162069, and GSE173905.

### 2.5. Immunohistochemistry Protein Level Expression of Eleven Markers

For the eleven selected markers, protein expression on ductal breast carcinoma was evaluated on the Protein Atlas server [30,31] at the following address: https://www.proteinatlas.org/ (accessed on 23 June 2023). Representative images were downloaded for each marker and quantification in tumor cells was estimated with its subcellular positivity.

### 2.6. Gene Expression Analyses and Association to the Breast Cancer Prognosis

Distant relapse-free survival (DRFS) from the GSE25066 dataset [17] was used as an outcome to performed iterative univariate Cox analysis against the expression of the genes identified as being significantly associated with the keyword “ferroptosis in breast”. During the trial follow-up of adjuvant therapy in breast cancer, distant relapse-free survival (DRFS) could be used as an endpoint [17]. DRFS was defined as the interval from initial diagnostic biopsy until diagnosis of distant metastasis or death due to breast cancer, non-breast cancer, or unknown causes [32]. This iteration of univariate Cox analysis was automatized with loopcolcox R-package version 1.0.0 available at the following address: https://github.com/cdesterke/loopcolcox (accessed on 18 May 2023). Univariate Kaplan–Meier and survival-optimal threshold on variables were performed with survminer R-package version 0.4.9 and survival R-package version 3.3.1. On genes with adverse prognosis association, functional enrichment was performed with the CPG signature from MsigDb database [33] through Toppgene online application [34]. A breast-cancer-related signature network was drawn with Cytoscape standalone software version 3.9.1 [35]. An expression molecular score related to ferroptosis/extracellular matrix remodeling functionalities was performed by computing the sum of the product between Cox beta-coefficients and expression of the eleven selected genes. For the eleven genes belonging to the ferroptosis/extracellular matrix remodeling signature, a multi-ROC analysis was performed against estrogen/progesterone receptor status detected in immunohistochemistry with the R-package multirocauc version 1.0.0 available at the address: https://github.com/cdesterke/multirocauc (accessed on 18 May 2023) (Appendix A). A multivariate Cox model was built with DRFS as the outcome and with incorporation of the expression molecular score and relevant clinical parameters. This DRFS multivariate model was assessed by testing the linearity of residuals at a global level and for each individual included parameters with Schoenfeld tests. Calibration of the DRFS multivariate model at 10 months of follow-up was carried out by 500 iterations of bootstrap with rms R package version 6.7.0. The nomogram validated at 10 months of follow-up was drawn for the DRFS multivariate model with regplot R package version 1.1.

## 3. Results

### 3.1. Ferroptosis Gene Expression Associated with the Prognosis of Patients with Breast Cancer

A text-mining approach was employed to identify ferroptosis-related genes in breast cancer literature. The scientific literature discussing genes, as stored in the MEDLINE database of biomedical references, has been used to prioritize genes based on the input supervised keywords to query the Pubmed database. The text-mining algorithm “Génie” was employed via querying Pubmed with “ferroptosis in breast” as the keywords. This query returned a list of 910 individual genes with a significant False Discovery Rate (FDR), and was positive in at least 10 distinct articles (Figure 1A): TP53 was found as the top ranked gene, followed by AKT1, EGFR, and HIF1A. Iterations of univariate Cox analyses against distant relapse-free survival (DRFS) outcomes in patients from the transcriptome dataset GSE25066 [17] was carried out for each of the 910 genes selected via text mining. For the best fifty ranked genes associated in their expression to the prognosis of patients, some were favorable and others adverse according to their beta-coefficients or hazard ratios (Figure 1B). 

Among the 252 significant genes associated with the prognosis, a filtration on the positivity of Cox beta-coefficients was carried out to retain 170 genes associated with adverse prognosis. This 170 adverse gene signature was significant in stratifying chemotherapy response prediction, such as dld-30 preoperative chemotherapy response prediction (Figure 2A, *p*-value = 7.48 × 10^−113^) [36] and neoadjuvant chemotherapy response by recurrent cancer burden (RBC) [37] (Figure 2B, *p*-value = 7.91 × 10^−8^). An expression molecular score was calculated with these 170 genes to verify its association with the DRFS of the patients. The optimal cutoff threshold was determined on this expression molecular score at a value of 687.02 (Appendix A). This score threshold identified 67 patients with a high score and 441 patients with a low score, and these two groups of patients were found with a significant difference in terms of DRFS prognosis. Patients with a high score had the worst prognosis and reached the DRFS median at 1.83 years (Appendix A). Functional enrichment performed with these 170 adverse genes on the MsigDb CPG signature database highlighted the major enrichment of these genes in breast cancer published signatures, such as SMID-breast cancer basal up [38] and SOTIRIOU-breast cancer grade 1 vs 3 up [39] (Figure 2C). A breast cancer network was drawn with ferroptosis, and a part of the ferroptosis-related genes were shared between the two independent breast cancer transcriptome cohort signatures (Figure 2D). These results suggest that the “ferroptosis in breast”-related text-mining approach is well adapted to explore breast cancer transcriptome cohorts according the prognosis of the patients.

To validate the Génie text-mining approach, an independent text-mining algorithm “GeneValorization” was employed to query the PUBMED database with “ferroptosis” as the keyword, but also some others with relevance in the context of the study, such as cancer stem cell (CSC), extracellular matrix (ECM) remodeling, breast cancer, lipid peroxidation, and regulated cell death. This validation allowed for highlighting the 15 top ranked-genes in text-mining sharing these keyword associations (Figure 3A): SET (SET nuclear proto-oncogene), TNF (tumor necrosis factor), HMOX1 (heme oxygenase 1), IL6 (interleukin 6), TRFC (transferrin receptor), ATF4 (activating transcription factor 4), HMGB1 (high mobility group box 1), KRAS (KRAS proto-oncogene, GTPase), EGFR (epidermal growth factor receptor), TLR4 (toll like receptor 4), HIF1A (hypoxia inducible factor 1 subunit alpha), ATG5 (autophagy-related 5), LCN2 (lipocalin 2), CDKN2A (cyclin dependent kinase inhibitor 2A), and MET (MET proto-oncogene, receptor tyrosine kinase). These fifteen best-ranked genes were verified having an adverse prognosis in the training cohort GSE25066 (Figure 3B).

### 3.2. Breast Cancer Eleven-Gene Signature Implicated in Ferroptosis and Extracellular Matrix Remodeling 

Among the fifteen best-ranked ferroptosis genes in breast cancer (Figure 3A), eleven harbored some association with “extracellular matrix remodeling” literature in the PUBMED database, especially TNF, SET, and IL6, found to be highly cited in this specific literature (Figure 4A). Querying MiPanda RNAseq server [40], an application that contains normalized transcriptome data from normal breast samples as the control versus primary and metastatic breast cancer samples, it could be observed that nine of these genes, except HIF1A and LCN2, were found significantly regulated in cancer samples (primary and metastatic) as compared to normal breast control (Appendix A). The 170 adverse ferroptosis-related genes were compared to the referent database of ferroptosis genes, such as FerrDB V2 [41] and genesets of KEGG [42] and Wikipathways [43]. This analysis highlighted that 8 of the 11 selected genes were also present in the FerrDB database, and a majority of them were categorized as ferroptosis drivers (KRAS, TLR4, HMGB1, EGFR, HIF1A, CDKN2A), except LCN2, which was categorized as a ferroptosis suppressor (Figure 4B), IL6 was categorized both as a driver and a suppressor. The prediction of the basal phenotype of breast cancer was tested against the expression of eleven genes via ROC analysis. CDKN2A expression was found to have the best aera under the curve (AUC), i.e., 0.83, followed by EGFR expression at 0.76 (Figure 4C). Contrastingly, for triple-negative breast cancer (TNBC), the phenotype EGFR was the best predictor at 0.78 followed by CDKN2A at 0.75 (Appendix A). 

The eleven-gene signature was evaluated also at a multi-omics level in the METABRIC validation cohort of breast cancer. The proportion of affected patients comprised between 4 and 7 percent of the patients for each gene, and there was concordance between alterations of each gene (Figure 5A). The maximumalterations were found in breast invasive ductal carcinoma and the minimumalterations were found in lobular carcinoma (Figure 5B). These alterations in the eleven-gene signature were present in patients with the worst relapse-free survival (Figure 5C) and significantly stratified patients based on major clinical parameters (Figure 5D) such as tumor histologic grade (Figure 5E) and Claudin-low/PAM50 classification (Figure 5F).

Through the Protein atlas server [30], the protein expression level of these eleven genes was assessed in the tissue section of breast ductal carcinoma. Each marker was confirmed as expressed in tumor cells at a protein level. Except HMGB1, the ten others were found expressed at the cytoplasm and membrane levels. Three of them were expressed at a nuclear level: HMGB1, SET, and HIF1A (Figure 6).

### 3.3. Ferroptosis/ECM Remodeling Signature Is Regulated by Ferroptosis Modifiers in Triple-Negative Breast Cancer Cells

To verify the link between ferroptosis function and gene members belonging to the eleven-gene signature, three distinct transcriptome datasets performed on MDA-MB-231 and HCC338 cells (cellular model of triple-negative breast cancer (TNBC)) were investigated. In the GSE173905 dataset, MDA-MB-231 cells were stimulated over 72 h with RSL3 and erastin [22]. Based on the expression of the eleven genes belonging to the ferroptosis/ECM molecular score, an unsupervised principal component analysis was performed with samples of GSE173905. This multivariate analysis confirmed that gene members composing the ferroptosis/ECM signature are regulated by the two distinct ferroptosis activators: erastin and RSL3 (*p*-value = 1.08 × 10^−5^, Figure 7A,B). The second dataset, GSE162069, comprised both in vitro stimulation and in vivo experiments. In the GSE162069 dataset, only eight of the eleven genes were quantified in these experiments (Figure 7C). For in vitro experiments, MDA-MB-231 were stimulated over 5 h with α-eleostearic acid (αESA) and ML162, two distinct glutathione peroxidase 4 (GPX4) inhibitors which induced ferroptosis [25]. In vitro, α-eleostearic acid (αESA) was found to be distinct from its control in principal component analysis (Figure 7D). Regarding in vivo experiments, an orthotopic xenograft was carried out in NSG mice for MDA-MB-231 cells, and the animals were treated orally with either 100 µL of safflower oil (control) or tung oil 5 days a week for 24 days [25]. In principal component analysis, in vivo treatment conditions were found to be distinct from their control (Figure 7D). In the third dataset, GSE154425, HCC38 cells were treated over 18 h with erastin (ferroptosis inducer), with or without tubacin, a HDAC6 inhibitor [28]. In this dataset, the eleven genes were mapped (Figure 7E). Upon principal component analysis, erastin had a specific action on the eleven-gene signature, which is not the case for tubacin. However, in combination, erastin action on the eleven-gene signature was modified by tubacin (Figure 7F).

### 3.4. Ferroptosis/ECM Remodeling Molecular Score Is an Independent Adverse Parameter in the Prognosis of Breast Cancer

Unsupervised principal component analysis based on the expression of the eleven ferroptosis/extracellular matrix remodeling-related genes were well stratified into groups of patients according to their histologic tumor grades (Figure 8A) but also their PAM50 subtype classification (Figure 8B) and triple-negative breast cancer (TNBC) phenotype (Figure 8C). This eleven-gene signature also revealed discriminant power for the prediction of preoperative chemotherapy based on the dld-30 classifier [36] (Figure 8D). Among the eleven genes, CDKN2A was the best predictor of the dld-30 classifier, with an AUC of 0.82, followed by EGFR with an AUC of 0.79 (Appendix A). For the prediction of residual breast cancer burden [37], the eleven-gene signature was inefficient (Appendix A). A molecular score was computed on the expression of the eleven ferroptosis/extracellular matrix remodeling-related genes. The optimal cutpoint was determined based on the DRFS residuals of the molecular score (Figure 8E). This threshold cutpoint was at 48.74 to stratify the breast cancer cohort in two groups. According the DRFS, the best age category stratification was found at 40.3 years old (Appendix A). No significant age difference was found between the two groups of patients harboring low and high values of the ferroptosis/ECM remodeling molecular score (*p*-value = 0.28, Table 1). Concerning the immunohistochemistry status of the estrogen receptor, a significantly higher proportion of negative patients was found in the group of patients with high values of the molecular score (*p*-value < 1 × 10^−4^, Table 1), and the observation was the same for progesterone receptor status (*p*-value < 1 × 10^−4^, Table 1). Concerning the TNBC phenotype, the group of patients with high values of the ferroptosis/ECM remodeling score presented a higher proportion of positive samples (*p*-value < 1 × 10^−4^, Table 1). Concerning pam50 molecular classification, the group of patients with high values of the ferroptosis/ECM remodeling score presented a higher proportion of basal type samples (*p*-value < 1 × 10^−4^, Table 1). No significant difference was observed on the tumor stages between the two groups of patients (*p*-value = 0.38, Table 1), but a higher proportion of patients N3 for the nodal status was observed in the group of patients with a high value of the molecular score (*p*-value = 0.013, Table 1). Concerning clinical AJCC staging, a significant difference was observed with an increasing proportion of stages IIIA, IIIB, and IIIC and inflammatory in the group of patients with a high value of the molecular score (*p*-value = 0.03, Table 1). As observed via PCA on the expression of the eleven genes which comprised the molecular score (Figure 4A), the grade variable was significant between the two groups of patients (*p*-value < 1 × 10^−4^, Table 1). The DRFS status and time parameters were also confirmed as significant between the two groups of patients (*p*-value < 1 × 10^−4^, Table 1). Effectively, Kaplan–Meier with the DRFS censor stratified based on ferroptosis/ECM remodeling was highly significant (Figure 8F), with the worst prognosis for patients who harbored a molecular score over 48.74 as the threshold, of which the median of the DRFS was 2.55 years. The type of taxanes (Taxol, Taxotere) administrated during the follow-up of the patients presented no associations with the ferroptosis/ECM remodeling groups of patients (*p*-value = 0.69, Table 1).

Relevant clinical parameters were integrated in a multivariable Cox model censored in the DRFS with group stratification based on the ferroptosis/ECM remodeling molecular score. This multivariable model, which harbored a concordant index of 0.77, was highly significant according to the likelihood ratio test (*p*-value = 2 × 10^−14^) (Figure 9A). The global and individual Schoenfeld test attested linear distribution residuals from the included parameters: age of patients, nodal status, pam50 molecular classification, grading, and ferroptosis/ECM remodeling molecular score (Appendix A). In this multivariable model, high values of the nodal status (N1 and N2,3) were found as independent adverse parameters of the DRFS (N1 versus N0 hazard ratio: 2.20, *p*-value = 5.91 × 10^−3^, N23 versus N0 hazard ratio: 3.39, *p*-value = 7.15 × 10^−5^, Table 2 and Figure 9A). Among molecular classification, the basal subtype appeared as an adverse group of breast cancer, with similar values to the reference (hazard ratio: 2.92, *p*-value 4.45 × 10^−2^, Table 2 and Figure 9A). In the DRFS multivariate model, the ferroptosis/ECM remodeling molecular score appeared as an adverse independent parameter in the prognosis of breast cancer patients (high score vs. low score hazard ratio: 2.69, *p*-value = 1.17 × 10^−5^). The multivariate model could be calibrated at ten months of follow-up with five hundred iterations using the Kaplan–Meier method (Figure 9B); this calibration showed that the multivariable model is stable at 10 months of follow-up. The corresponding nomogram of the model was drawn for a prediction at 10 months of follow-up (Figure 9C). This representation confirms the important part of the molecular score in the multivariate model. Indeed, at 10 months of follow-up, the molecular score appears dispersed between the range of point values of the model (10–70), as the DRFS probability at 10 months was between 0.006 and 0.1 (Figure 9C).

## 4. Discussion

During this study, a list of genes was defined using the text-mining approach related to ferroptosis cellular functionality known as an important way of cellular death implicated in tumor response to therapies [10]. Surprisingly, in the transcriptome of breast tumors under therapies (GSE173905) [17], the majority of ferroptosis-related genes presented expression associated with adverse distant relapse-free survival, whereby 170 of 252 significant genes were found to have an univariate hazard ratio over 1 (Appendix A). Triple-negative breast cancer (TNBC) is the breast cancer subtype with the worst prognosis, and it has a strong invasive and metastatic capacity and easily invades into blood vessels, thus increasing the recurrence rate [44]. Due to the lack of ER, PR, and HER2 receptor expression, therapeutic methods for TNBC are much more limited compared with other breast cancer types. Ferroptosis is a modality of regulated cell death driven by iron-dependent lipid peroxidation [6] and TNBC cells are sensitive to ferroptosis inducers [45,46], suggesting this new form of non-apoptotic cell death as an attractive target for the treatment of the “difficult-to-treat” tumor [47].

TNBC is a heterogeneous disease which has been divided by transcriptome analyses in seven TNBC subtypes: basal-like 1 (BL1), basal-like 2 (BL2), immunomodulatory (IM), mesenchymal (M), mesenchymal stem-like, luminal androgen receptor, and unclassified (UNS), with distinct proportions of mesenchymal remodeling, immune infiltration, or androgen receptor expression between subtypes [48]. According to the expression level of GPX4, a heterogenous response of TNBC was observed to ferroptosis therapy (GPX4 inhibitor) in combination to immunotherapy with a better therapy response for the LAR TNBC subtype [49]. 

With an independent text mining application querying the NCBI database, GeneValorization [15], a Ferroptosis/ECM remodeling molecular score in basal breast cancer [17] has been established based on the expression of eleven related genes, i.e., TNF, IL6, SET, CDKN2A, EGFR, HMGB1, KRAS, MET, LCN2, HIF1A, and TLR4. These molecules have been verified to be regulated by distinct ferroptosis inducers in TNBC cellular models [8,9,25,28] and expressed at a protein level in ductal breast carcinoma tissue sections. 

For the majority of the genes contained in the eleven-gene signature, it could be possible to link the literature individually to the ferroptosis and ECM remodeling context.

CDKN2A is frequently deleted by the DNA copy number variation analysis in luminal androgen receptor (LAR) TNBC subtype [50]. During glioblastoma (GBM), CDKN2A deletion remodels the GBM lipidome, notably redistributing oxidizable polyunsaturated fatty acids into distinct lipid compartments, and CDKN2A-deleted GBMs display higher lipid peroxidation, selectively priming tumors for ferroptosis [51]. Cancer-associated fibroblasts (CAFs), the most abundant and likely active cellular component of breast cancer-associated stroma, promote carcinogenesis through paracrine effects. During breast cancer, CDKN2A expression is reduced in 83% of cancer-associated fibroblasts as compared with their normal adjacent cancer-free counterpart tissues isolated from the same patients. CDKN2A downregulation using specific siRNA activated breast fibroblasts and increased the expression/secretion levels of stromal-cell-derived factor 1 (SDF-1) and matrix metalloproteinase (MMP)-2 [52].

HIF-1α is an important regulator of lipid metabolism [53]. Hypoxia-induced lipid metabolism reprogramming results in fatty acid accumulation, which promotes tumor growth and survival upon reoxidation [54]. HIF1A is a negative regulator of erastin- or RSL3-induced ferroptosis in human fibrosarcoma HT1080 and non-small cell lung cancer Calu-1 cells, and this anti-ferroptosis effect is linked to the activation of clockophagy, a type of selective autophagy for the degradation of the core circadian clock protein, ARNTL [55,56]. In the context of increased hypoxia/HIF1A and ECM stiffness in chemoresistant tumors, a high expression of HIF1A could be adverse because it leads to the upregulation of ITGA5, activation of the downstream FAK/Src signaling pathways, and repression of miR-326, which targets fibronectin (FN1), an extracellular matrix (ECM) central chemoresistance driver gene [57]. 

Adipokine lipocalin-2 (LCN2) has been demonstrated to be an ECM regulator through its association with the ECM protease matrix metalloproteinase-9 (MMP-9) [58]. It has been shown that LCN2 knockout in the human breast cancer cell line MDA-MB-231 ameliorates erastin-mediated ferroptosis and increases cisplatin vulnerability [59].

Adipocytes constitute the main cell component of the ECM in breast cancer [60]. Cancer-associated adipocytes (CAAs) are localized at the invasive front of breast tumor and exhibit a modified phenotype, loss of lipid content, decrease in late adipocyte differentiation markers, and overexpression of inflammatory cytokines and proteases [61]. In breast tumors, IL6 is secreted via CAAs, which play essential roles in favor of proliferation, angiogenesis, dissemination, invasion, and metastasis of breast cancer [62], and its production is associated with therapy resistance [63]. Tumor-associated macrophages (TAMs) are major components of the tumor microenvironment (TME), which are closely associated with the tumor malignant progression. In TNBC, hepatic leukemia factor (HLF) transactivated gamma-glutamyltransferase 1 (GGT1) promote the ferroptosis resistance and interactive dialogue between TNBC cells, and TAMs promotes sustained activation of HLF in tumor cells through the IL-6–TGF-β1 axis [64].

EGFR promoted TNBC cell clustering, and the blockade of EGFR successfully abolished tumor cell cluster formation [65]. It has been shown that inhibition of the EGFR signaling pathway significantly suppressed cell viability of TNBC cells and reduced the fraction of CSCs with intracellular enhancement of lipid peroxidation when TNBC cells are exposed to erastin [66]. The increased metastatic potential of TNBC is a combined result of an extensive extracellular matrix (ECM) remodeling that leads to cytoskeleton rearrangement and activation of epithelial-to-mesenchymal transition (EMT). The overexpression of epidermal growth factor receptor (EGFR) in TNBC tumors has been linked to an induced expression of EMT-related molecules [67].

MET is known to be implicated in chemotherapy resistance, including those targeting EGFR, BRAF, and MEK, but also contributes to cytotoxic chemotherapy resistance [68]. Its ligand, HGF, is a pleiotropic factor produced by mesenchymal cells in the stroma, and as such, it is widely distributed in the extracellular matrix of most tissues [69]. Dysregulation of the MET/HGF pathway leads to uncontrolled cell proliferation and oncogenesis, and is observed in multiple tumor types [70]. HGF is known to exacerbate pancreatic cancer cell ferroptosis resistance [71]. 

HMGB1 is implicated in regulating stress responses to oxidative damage and cell death, and can be released into the extracellular space to act as a damage-associated molecular pattern protein during ferroptosis [72]. HMGB1 is known to act via the NRF2/GPX4 axis to repress ferroptosis in mesangial cells in response to high glucose [73]. In TNBC, the downregulation of miR-205 contributes to epithelial–mesenchymal transition and invasion of cancer cells by targeting the HMGB1-RAGE signaling pathway [74].

KRAS mutations are known as very infrequent in triple-negative breast tumors [75], but in basal breast cancer, KRAS has been shown to promote the mesenchymal features of this aggressive cancer [76]. In the tumor microenvironment, tumor-associated macrophage polarization could be driven by ferroptosis via the release and uptake of the oncogenic KRAS protein [77].

The overexpression of Toll-like receptor-4 (TLR4) in human tumors often correlates with chemoresistance and metastasis. The depletion of TLR4 in naturally overexpressing MDA-MB-231 cells downregulated prosurvival genes concomitant with two- to three-fold reduced IC(50) to paclitaxel in vitro and a six-fold decrease in the recurrence rate in vivo [78]. The role of TLR4 in ferroptosis has been demonstrated in the hippocampal hypoxic-ischemic context [79] and in renal ischemia [80].

The SET nuclear proto-oncogene is known to be upregulated in TNBC tumor samples with CIP2A. Ectopic expression of SET in MDA-MB-231 and MDA-MB-468 increased pAkt, pERK, pElk-1, and CIP2A expressions. The use of a protein–protein binding antagonist (TD19) between SET and PP2A induced the downregulation of CIP2A through ERK phosphorylation and downstream nuclear translocation of Elk-1, suggesting a molecular regulation between SET and CIP2A via the MAPK pathway. Targeting SET to disrupt the oncogenic CIP2A loop could be a promising TNBC therapy [81]. No evidence of relation between the SET nuclear proto-oncogene and ferroptosis in the literature was found, but our work showed (Figure 7) a regulation of SET nuclear proto-oncogene under ferroptosis inducers in MDA-MB-231 and HCC338 TNBC cancer cells.

The activities of cancer-associated fibroblasts (CAFs) and mesenchymal stromal cells (MSCs) in breast cancer are integrated within an intimate inflammatory tumor microenvironment (TME) that includes high levels of tumor necrosis factor α (TNF-α). During the in vitro conversion process of mesenchymal stromal cells in cancer-associated fibroblast by breast tumor cell (MDA-MB-231 and MCF-7)-conditioned media, TNF-alpha stimulation is responsible for the chemokines released (CCL2, CXCL8, and CCL5) by the tumor stromal cells [82]. During cancer immunostimulation, the secretion of TNF downregulates the expression of SLC7A11 and SLC3A2, and reduces the absorption of cysteine, leading to lipid peroxidation and iron deposition in cancer cells [83].

## 5. Conclusions and Perspectives

In the present work, the expression of the genes associated with bad breast cancer prognosis was investigated employing a text-mining approach and transcriptome data integration of relationships between ferroptosis and ECM remodeling functions. This adverse regulated program allowed for computing a molecular expression score that could be promising to evaluate the response to ferroptosis target therapies in breast cancer.

## Figures and Tables

**Figure 1 cells-12-02176-f001:**
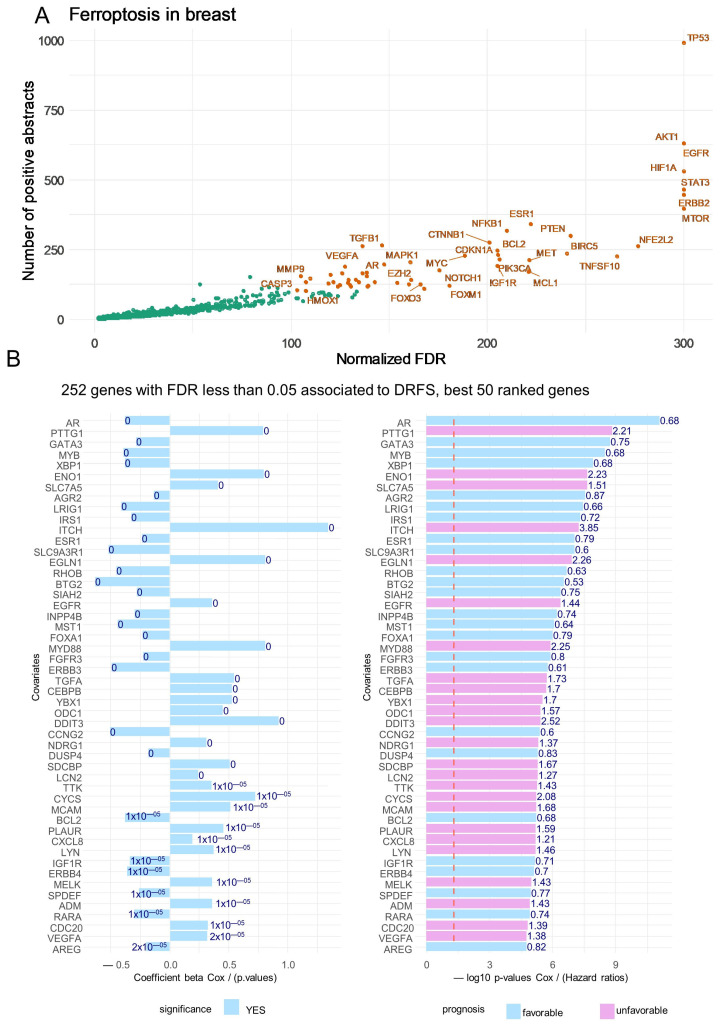
Gene expression profile related to ferroptosis functionality is associated with disease-free relapse survival in breast cancer. (**A**) Scatterplot of the text-mining normalized false discovery rate (FDR): negative log10 q-values versus number of positive articles in Pubmed for genes related to ferroptosis in breast cancer (green dots correspond to less significant selected genes); (**B**) Barplots of univariate Cox beta-coefficients and negative log10 *p*-values for the 50 best ferroptosis-related genes according disease-free relapse survival (DFRS) of breast cancer patients (transcriptome GSE25066, n = 508).

**Figure 2 cells-12-02176-f002:**
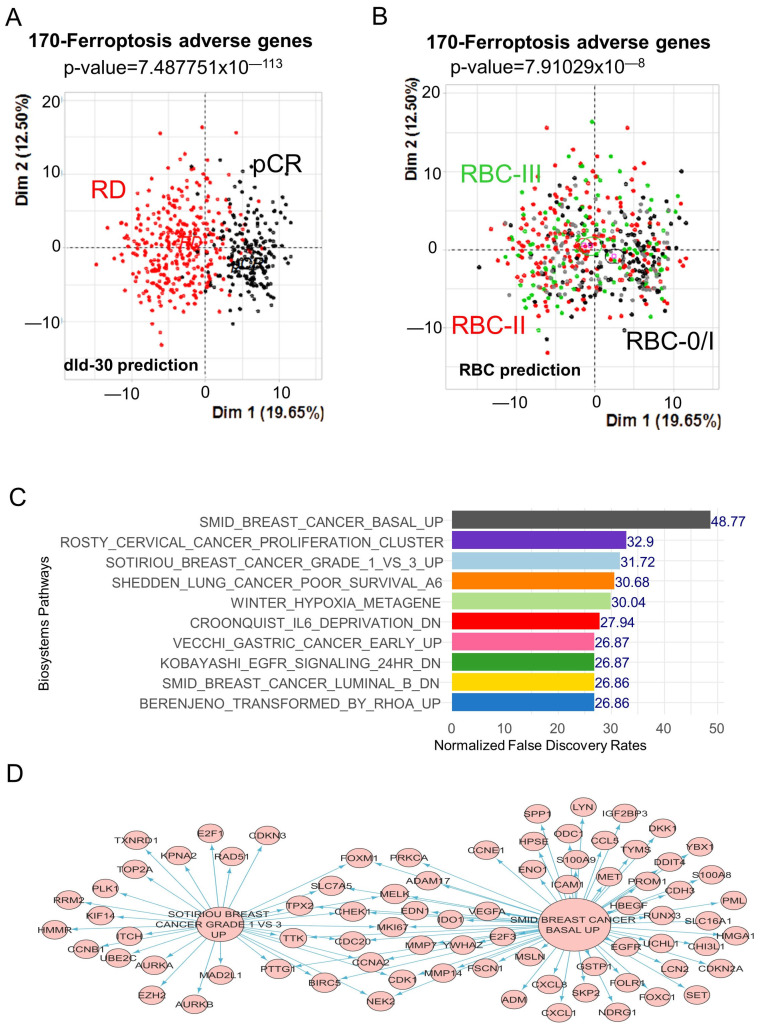
Unfavorable ferroptosis-related genes were enriched in basal breast cancer. (**A**) Principal component analysis based on the expression of 170 adverse ferroptosis genes and stratified on dld-30 response prediction. (**B**) Principal component analysis based on expression of the 170 adverse ferroptosis genes and stratified by residual cancer burden response prediction. (**C**) Barplot of functional enrichment performed on the “MSIGDB CPG” database with the 170 unfavorable ferroptosis genes. (**D**) Functional enrichment network of 170 unfavorable ferroptosis genes enriched in advanced breast cancer signatures.

**Figure 3 cells-12-02176-f003:**
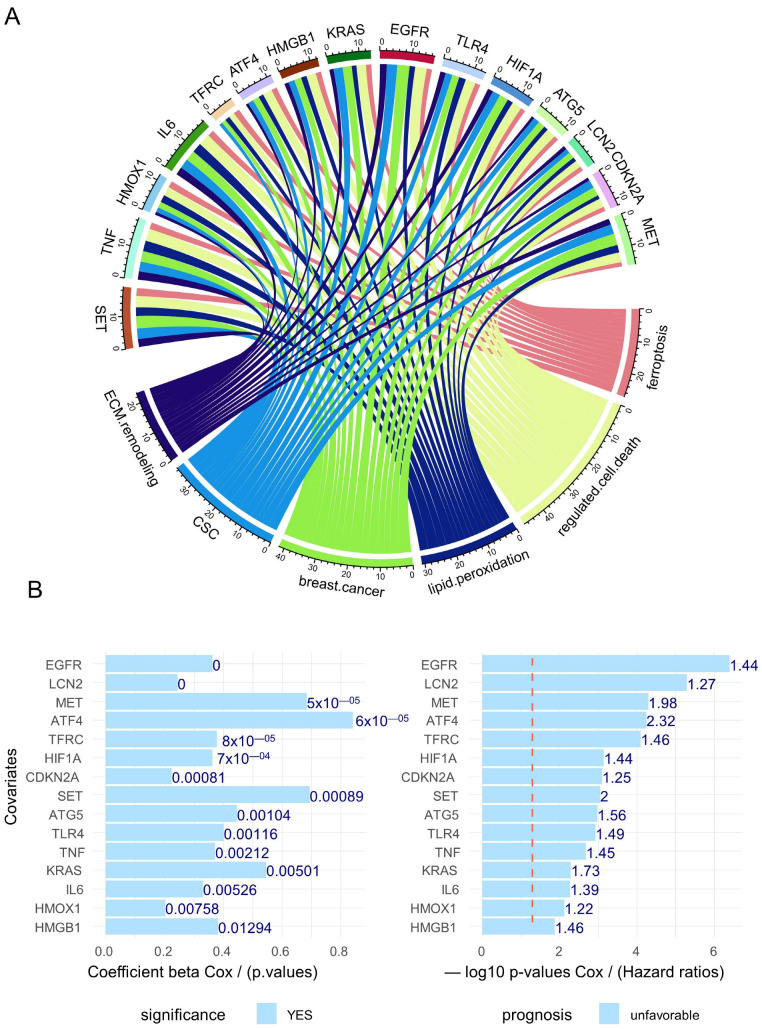
Top fifteen genes prioritized by text mining and having adverse prognosis. (**A**) Circosplot of the 15 best ferroptosis genes most cited in literature associated with the following keywords: extracellular matrix (ECM) remodeling, cancer stem cell (CSC), breast cancer, lipid peroxidation, regulated cell death, and ferroptosis. (**B**) Barplot of Cox analyses of the top15 gene with DRFS (distant relapse-free survival) as the outcome.

**Figure 4 cells-12-02176-f004:**
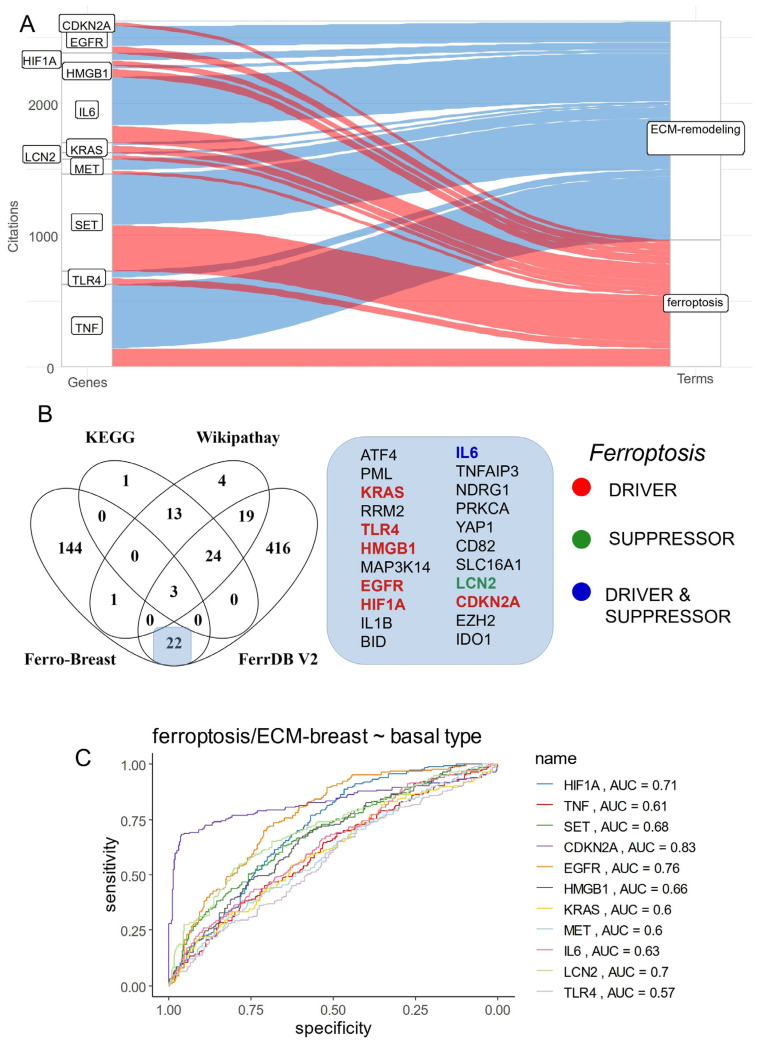
Eleven-gene signature shared between ferroptosis and extracellular matrix remodeling functionalities. (**A**) Alluvial plot of literature citation counts for the best 11 genes with co-occurrence in ferroptosis and extracellular matrix remodeling (ECM) functionalities. (**B**) Venn diagram testing overlap between 170 adverse gene signature and the ferroptosis databases. (**C**) Multi-ROC analysis of the expression (GSE25066) for the 11 genes against basal phenotype.

**Figure 5 cells-12-02176-f005:**
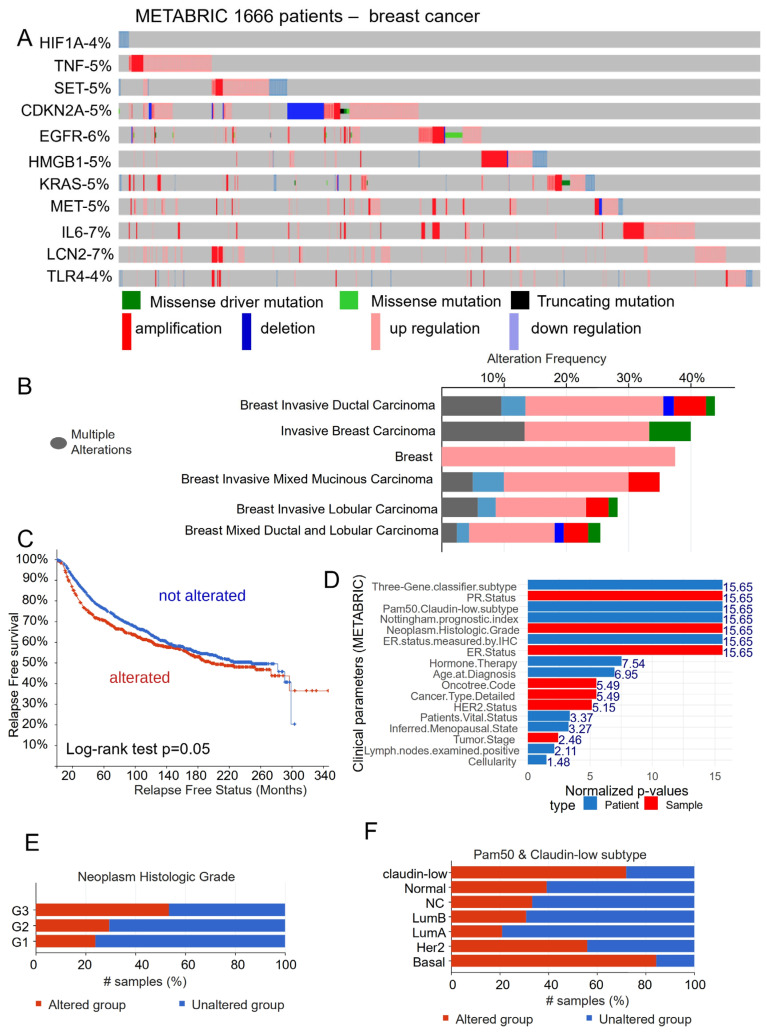
Validation of the eleven-gene signature in the METBRIC breast cancer cohort. (**A**) Oncoprint of alterations affecting the eleven genes. (**B**) Barplot of the alteration frequencies by subtypes of breast cancer. (**C**) Relapse-free survival analysis stratified based on alterations. (**D**) Barplot of clinical parameters associated with the eleven gene alterations. (**E**) Example of association with tumor histologic grade. (**F**) Example of association with claudin-low/PAM50 classification.

**Figure 6 cells-12-02176-f006:**
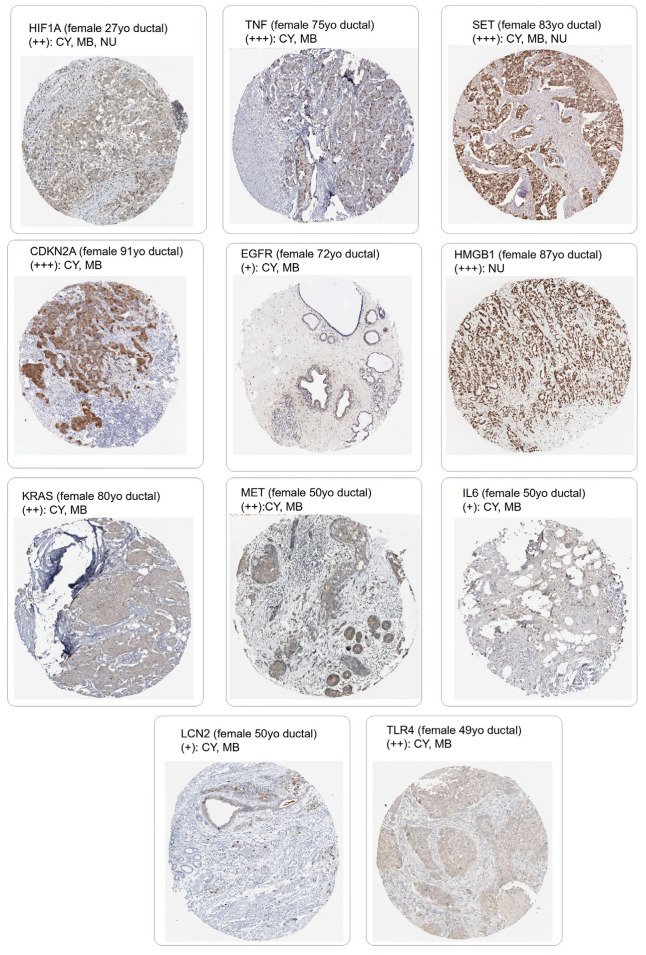
Immunohistochemistry of ductal breast carcinoma tissue section for testing the eleven markers of the ferroptosis/ECM signature. Representative expression image section was extracted from the Protein Atlas server, tumor cell expression was quantified according three levels of the cross (tumor cell staining intensities: +++, strong; ++, moderate; +, weak), and subcellular localization in the tumor cell was annotated as follows: NU: nuclear, CY: cytoplasm, MB: membrane.

**Figure 7 cells-12-02176-f007:**
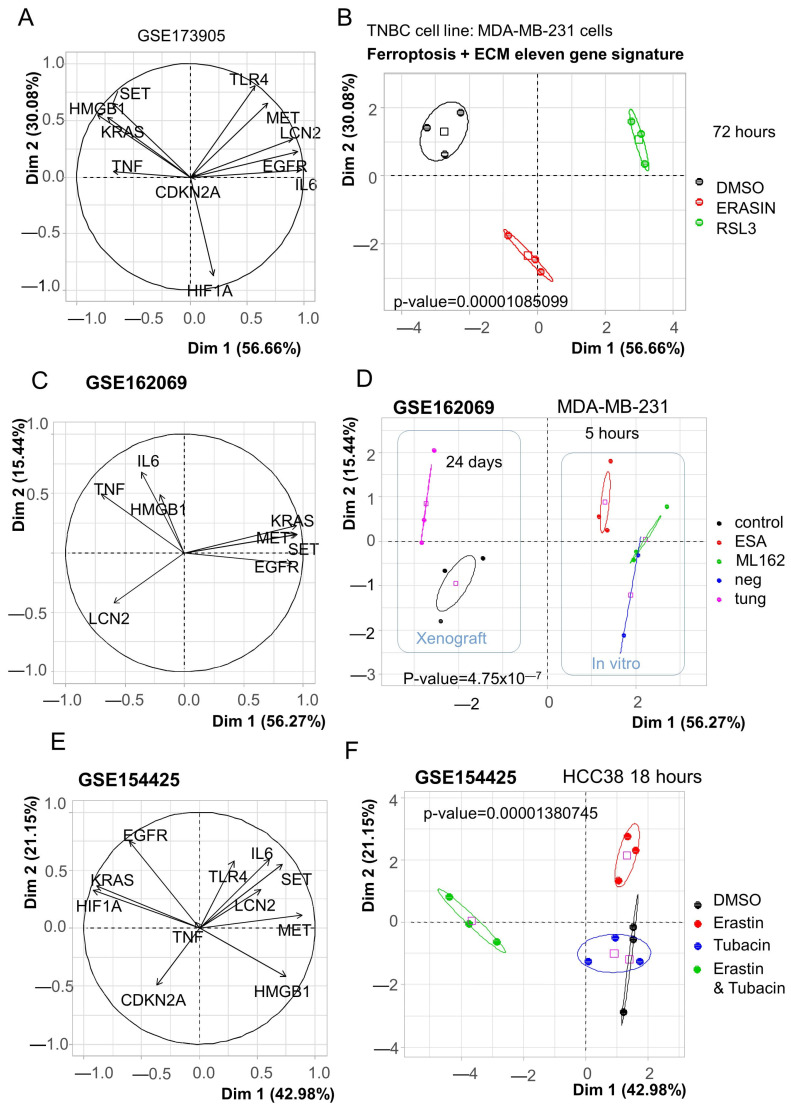
Regulation of the ferroptosis/extracellular matrix remodeling signature by ferroptosis inducers in triple-negative breast cancer cellular models. On principal component plots, the pink colored square symbols represent the barycenters of the groups. (**A**) Principal component correlation plot for the dataset GSE173905. (**B**) First principal map for the dataset GSE173905. (**C**) Principal component correlation plot for the dataset GSE162069. (**D**) First principal map for the dataset GSE154425 (in vitro and in vivo). (**E**) Principal component correlation plot for the dataset GSE162069. (**F**) First principal map for the dataset GSE154425.

**Figure 8 cells-12-02176-f008:**
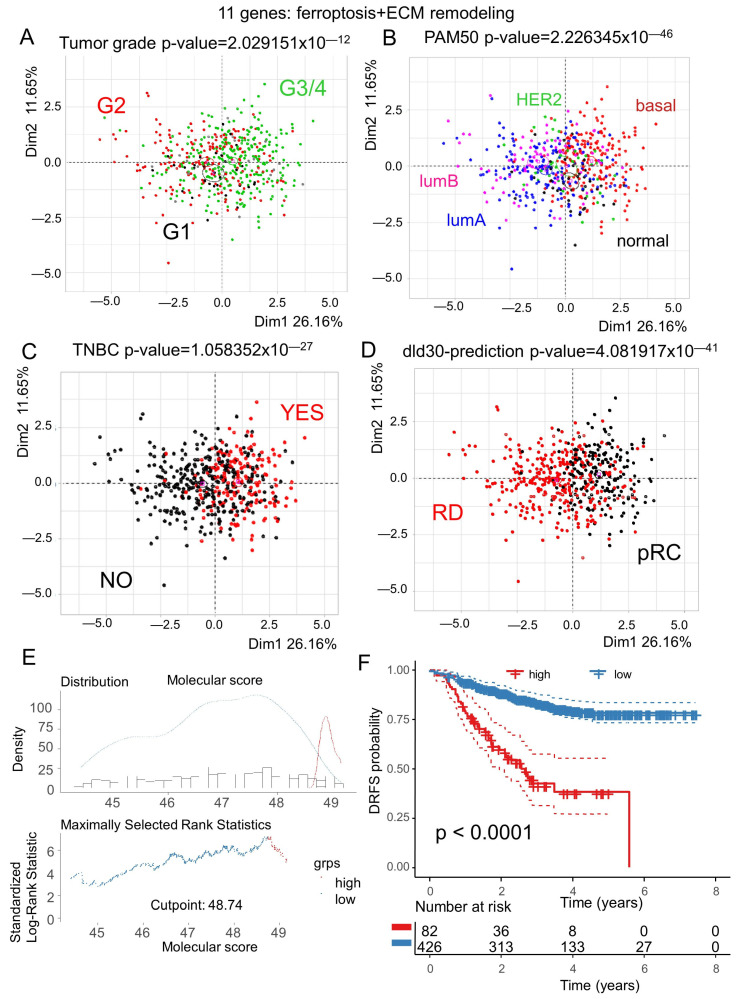
Ferroptosis/extracellular matrix remodeling molecular score is associated with a worse prognosis in breast cancer. For the transcriptome dataset (GSE25066), unsupervised principal component analysis was performed with the eleven-gene signature and stratified based on: (**A**) tumor grades (grades 3 and 4 were aggregated in one class), (**B**) pam50 molecular classification of breast tumors, (**C**) TNBC phenotype, and (**D**) dld-30 preoperative chemotherapy response. (**E**) Optimal threshold cutpoint determined for ferroptosis/ECM remodeling molecular score censored on the DRFS (distant relapse-free survival). (**F**) Kaplan–Meier and log-rank analyses censored on the DRFS and stratified based on the ferroptosis/ECM remodeling molecular score threshold.

**Figure 9 cells-12-02176-f009:**
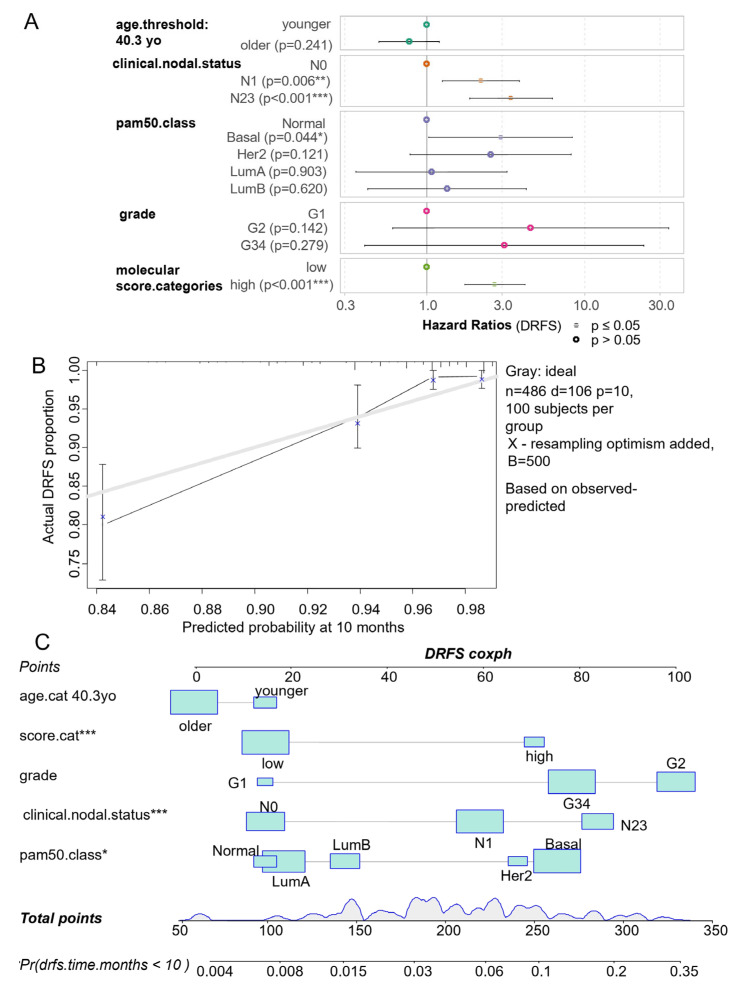
Ferroptosis/extracellular matrix remodeling molecular score is an independent adverse parameter in the prognosis of breast cancer patients. (**A**) Forestplot of the multivariable model censored based on the distant relapse-free survival, including the ferroptosis/extracellular matrix remodeling and clinico-biological relevant parameters, such as age, nodular status, grade, and molecule classes; significance: *: 0.01 < *p* < 0.05, **: 0.001 < *p* < 0.01, ***: *p* < 0.001. (**B**) Bootstrap calibration plot of the DRFS multivariable model performed with 500 iterations using the Kaplan–Meier method at 10 months of follow-up: grey line (optimal model). (**C**) Nomogram of the DRFS multivariable model predicted at 10 months of follow-up.

**Table 1 cells-12-02176-t001:** Breast cancer cohort from the GSE25066 dataset stratified between high and low levels of the ferroptosis/extracellular matrix remodeling molecular score.

Variable	Level	Low (n = 426)	High (n = 82)	Total (n = 508)	*p*-Value
age.years	mean (sd)	49.7 (10.4)	50.5 (10.8)	49.8 (10.5)	0.524161
age.categories (40.3 yo)	younger	79 (18.5)	20 (24.4)	99 (19.5)	
	older	347 (81.5)	62 (75.6)	409 (80.5)	0.283927
er.status.ihc	Negative	144 (34.1)	61 (76.2)	205 (40.8)	
	Positive	278 (65.9)	19 (23.8)	297 (59.2)	<0.0001
	missing	4	2	6	
pr.status.ihc	Negative	192 (45.6)	66 (82.5)	258 (51.5)	
	Positive	229 (54.4)	14 (17.5)	243 (48.5)	<0.0001
	missing	5	2	7	
TNBC	no	289 (70.3)	22 (28.2)	311 (63.6)	
	YES	122 (29.7)	56 (71.8)	178 (36.4)	<0.0001
	missing	15	4	19	
pam50.class	Normal	42 (9.9)	2 (2.4)	44 (8.7)	
	Basal	123 (28.9)	66 (80.5)	189 (37.2)	
	Her2	33 (7.7)	4 (4.9)	37 (7.3)	
	LumA	152 (35.7)	8 (9.8)	160 (31.5)	
	LumB	76 (17.8)	2 (2.4)	78 (15.4)	<0.0001
clinical.tumor.stage	T-0,1,2	247 (58.0)	41 (50.0)	288 (56.7)	
	T-3	119 (27.9)	26 (31.7)	145 (28.5)	
	T-4	60 (14.1)	15 (18.3)	75 (14.8)	0.379010
clinical.nodal.status	N-0	140 (32.9)	17 (20.7)	157 (30.9)	
	N-1	205 (48.1)	39 (47.6)	244 (48.0)	
	N-2,3	81 (19.0)	26 (31.7)	107 (21.1)	0.013986
clinical.ajcc.stage	IIB	131 (30.8)	20 (24.4)	151 (29.7)	
	IIIA	99 (23.2)	22 (26.8)	121 (23.8)	
	IIIB	63 (14.8)	17 (20.7)	80 (15.7)	
	IIA	109 (25.6)	12 (14.6)	121 (23.8)	
	IIIC	16 (3.8)	7 (8.5)	23 (4.5)	
	Inflammatory	2 (0.5)	2 (2.4)	4 (0.8)	
	I	6 (1.4)	2 (2.4)	8 (1.6)	0.033976
grade	G-1	32 (7.8)	0 (0.0)	32 (6.6)	
	G-2	167 (40.8)	13 (16.9)	180 (37.0)	
	G-3,4	210 (51.3)	64 (83.1)	274 (56.4)	<0.0001
	missing	17	5	22	
drfs.status	1	70 (16.4)	41 (50.0)	111 (21.9)	
	0	356 (83.6)	41 (50.0)	397 (78.1)	<0.0001
drfs.time.years	mean (sd)	3.2 (1.6)	2 (1.2)	3 (1.6)	<0.0001
type.taxane	Taxotere	78 (45.6)	14 (51.9)	92 (46.5)	
	Taxol	93 (54.4)	13 (48.1)	106 (53.5)	0.691854
	missing	255	55	310	

**Table 2 cells-12-02176-t002:** Disease-free relapse survival multivariate model including the ferroptosis/extracellular matrix remodeling molecular score.

Variables	Hazard Ratios	Confidence-Low	Confidence-High	*p*-Value
age.cat older	0.770	0.498	1.192	2.41 × 10^−1^
clinical.nodal.statusN1	2.202	1.255	3.862	5.91 × 10^−3^
clinical.nodal.statusN23	3.395	1.857	6.205	7.15 × 10^−5^
pam50.classBasal	2.924	1.027	8.327	4.45 × 10^−2^
pam50.classHer2	2.530	0.783	8.177	1.21 × 10^−1^
pam50.classLumA	1.071	0.356	3.221	9.03 × 10^−1^
pam50.classLumB	1.339	0.422	4.252	6.20 × 10^−1^
Grade.G2	4.526	0.603	33.950	1.42 × 10^−1^
Grade.G34	3.085	0.402	23.693	2.79 × 10^−1^
score.high	2.689	1.728	4.185	1.17 × 10^−5^

## Data Availability

All original sources have been appropriately referenced.

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
