# Peer review of "Adverse Crosstalk between Extracellular Matrix Remodeling and Ferroptosis in Basal Breast Cancer"

_cells, 2023, doi:10.3390/cells12172176_

Round 1
Reviewer 1 Report
In this paper, the authors used literature text-mining and transcriptomic data in breast cancer and identified 910 “ferroptosis in breast” genes with significant FDR. They then selected 252 genes based on their hazard ratio (HR) using COX analysis in breast cancer, and further filtered these 252 genes and identified 171 genes, which were associated to adverse prognosis in breast cancer patients. Among the 171 genes, they identified 11 genes that were associated with extracellular matrix (ECM) literature. Accordingly, the authors concluded that the described approach enabled them to identify eleven predictive genes of worst prognosis in breast cancer and crosstalk between ferroptosis and ECM remodeling.
While the study introduces an interesting approach using literature text-mining to link between ECM remodeling and ferroptosis and identified 11 possible prognostic genes in breast cancer, the analysis is very limited to a few selected datasets and in a way biased, and therefore the conclusions of this study are not convincing enough.
Major comments:
(1) The authors have selected genes by text mining, using the term “ferroptosis in breast” to pick 910 genes. Then, they use Cox model to calculate the hazard ratio (HR) for each gene. It is not clear whether there is a connection between the HR and the text mining results. Some of the genes shown in Fig. 1B are less relevant to ferroptosis than others (such as BCL2), suggesting that another step of gene filtration, following the text-mining, might be beneficial (see for example, similar works done on ferroptosis-related genes taken from FerrDB: doi.org/10.3389/fgene.2021.758981).
(2) The authors can broaden the discussion on whether the HR is different between pro- or anti- ferroptotic genes. Specifically, it would be important to know how many of the 171 genes selected with positive HR are bonna-fide ferroptosis related (for example, by comparing to FerrDB). Also, since the authors quantify the enrichment of MsigDB CPG in those 171 genes, they can also examine the enrichment of KEGG_ferroptosis or WP_ferroptosis which are also part of the MsigDB [c2 set].
(3) The meaning of ROC curves shown in Fig. 3 is unclear. If the author suggest that CDKN2A can be used to predict ER or PR statues, it is unclear how is might be relevant to ferroptosis? We predict that other genes (not related to ferroptosis), would have higher AUCs. The choice of the clinical parameters tested in those ROC curves appear to be arbitrary. Using clinical data more relevant to ferroptosis might be in order here. For example, the authors can use PAM50 (for basal/mesenchymal differentiation, which is relevant to ferroptosis), or correlate their signature to the expression levels of genes which might affect iron levels, etc.
(4) According to the text, the authors validated the 11-gene ferroptosis/ECM signature in MDA-MB-231 cells treated with erastin and RSL3. However, the dataset used, GSE173905, contains gene expression at 72 hr post treatment of the surviving cells.
Hence, many genes of Fig. 6 cannot be considered as response gene expression but rather may play a compensatory role to prevent ferroptosis and protect the cells from death response. The text must highlight this fact, including sentences from the discussion such as “..it was observed that RSL3 .. induced overexpression of MET in MDA-MB-231”. The authors can compare their results to datasets such as GSE162069, GSE154425, in which TNBC cells were treated with ferroptosis inducers for shorter time points (therefore showing gene expression in response to the inducers).
(5) It is also unclear why the authors choose the GSE173905 dataset to validate the 11 genes, since Figure 6C-E makes it clear that there is no relation between the HR of the 11 genes and their expression in this dataset. The authors didn’t discuss this discrepancy in the discussion.
(6) The 11 genes were chosen based on their positive HR in one dataset. It is very important to validate it in another breast cancer patients datasets, such as METABRIC, which has very good prognostic data
(https://www.cbioportal.org/study/summary?id=brca_metabric).
Minor comments:
(1) Figure 1A: The authors show a qqplot to display the text mining results. While the distribution of p-values is interesting, other important parameters can be revealed by using another type of plot, for example a scatterplot showing the p-values vs. the number of articles. It might also be beneficial to label the genes with the highest p-values.
(2) Figure 2C: this analysis is very interesting, however, it might be beneficial to mention the hazard ratios of these 15 genes, to connect these genes to the results shown in 1B.
(3) Figure 4D: Since the 11-gene score was done using the HR values of the genes, the KM plot is unsurprisingly significant. Therefore, it will be interesting to know how this KM, based on the 11 ferroptosis/ECM genes, is compared to a KM plot made using all the 171 genes. Does focusing only on the 11 ECM related genes improves the result over the 171 genes?
(4) Figure 4/table 1 – since the authors later work on TNBC models, it will be interesting to show the 11-gene scores in TNBC vs. non-TNBC in Fig. 4 and table 1 (TNBC data can be easily deduced from the clinical data).
(5) The dataset used, GSE25066, contains other transcriptomic based prediction (DLDA-30), as well as residual cancer burden (RCB) prediction. It would be very interesting to compare the authors’ predictions (using the 171 genes or 11 ferroptosis/ECM genes) with these predictions, to increase the impact of the authors signatures.
Author Response
In this paper, the authors used literature text-mining and transcriptomic data in breast cancer and identified 910 “ferroptosis in breast” genes with significant FDR. They then selected 252 genes based on their hazard ratio (HR) using COX analysis in breast cancer, and further filtered these 252 genes and identified 171 genes, which were associated to adverse prognosis in breast cancer patients. Among the 171 genes, they identified 11 genes that were associated with extracellular matrix (ECM) literature. Accordingly, the authors concluded that the described approach enabled them to identify eleven predictive genes of worst prognosis in breast cancer and crosstalk between ferroptosis and ECM remodeling.
Answer: Sorry for the error in the previous version, 171 genes were reported but only 170 were significant: the header of column was taken in account in the list quantification. This error was rectified in the present R1 version.
While the study introduces an interesting approach using literature text-mining to link between ECM remodeling and ferroptosis and identified 11 possible prognostic genes in breast cancer, the analysis is very limited to a few selected datasets and in a way biased, and therefore the conclusions of this study are not convincing enough.
Answer: Thank you for your comment: METABRIC cohort at multi-omics level was investigated to confirm this signature as you suggested.
Major comments:
(1) The authors have selected genes by text mining, using the term “ferroptosis in breast” to pick 910 genes. Then, they use Cox model to calculate the hazard ratio (HR) for each gene. It is not clear whether there is a connection between the HR and the text mining results. Some of the genes shown in Fig. 1B are less relevant to ferroptosis than others (such as BCL2), suggesting that another step of gene filtration, following the text-mining, might be beneficial (see for example, similar works done on ferroptosis-related genes taken from FerrDB: doi.org/10.3389/fgene.2021.758981).
Answer: We did not take the decision to use the ferroptosis FerrDB in prognostication of the breast cancer because this job was already published in the following paper: https://pubmed.ncbi.nlm.nih.gov/34222241/ in which the authors used FerrDB database for cox analyses on TCGA BREAST cancer cohort. So, it is for this main reason we employed an original text-mining approach to identify new relations between “ferroptosis in breast” and coding genes in the scientific and medical literature uploaded on NCBI Pubmed database.
By comparing our final eleven gene “Ferroptosis-ECM in breast” signature with FerrDB database V2, we observed that 8 of the eleven selected genes in our signature were also present in FerrDB database V2 suggesting that the majority of genes from our in-house signature is mostly validated by referent FerrDB database. We could see an originality of our text mining approach corresponding to: TNF, MET, SET genes enriched in the present work and not present in concurrent databases such as: FerrDB, Wikipathway and KEGG. (See the added new Venn diagram)
(2) The authors can broaden the discussion on whether the HR is different between pro- or anti- ferroptotic genes. Specifically, it would be important to know how many of the 170 genes selected with positive HR are bonna-fide ferroptosis related (for example, by comparing to FerrDB). Also, since the authors quantify the enrichment of MsigDB CPG in those 170 genes, they can also examine the enrichment of KEGG_ferroptosis or WP_ferroptosis which are also part of the MsigDB [c2 set].
Answer: Thank you for this good idea, in Venn diagram we see that 8 of our 11 genes were found common with FerrDB V2, so we could comment sub-types of ferroptosis action for these last ones: majority of them were found as only “DRIVER” of ferroptosis (HIF1A, CDKN2A, KRAS, EGFR, TLR4, HMGB1), one was found as only “SUPPRESSOR” of ferroptosis (LCN2) and one of them was found as both “DRIVER” and “SUPPRESSOR” of ferroptosis (IL6). These annotations were put on the new Venn diagram.
(3) The meaning of ROC curves shown in Fig. 3 is unclear. If the author suggest that CDKN2A can be used to predict ER or PR statues, it is unclear how is might be relevant to ferroptosis? We predict that other genes (not related to ferroptosis), would have higher AUCs. The choice of the clinical parameters tested in those ROC curves appear to be arbitrary. Using clinical data more relevant to ferroptosis might be in order here. For example, the authors can use PAM50 (for basal/mesenchymal differentiation, which is relevant to ferroptosis), or correlate their signature to the expression levels of genes which might affect iron levels, etc.
Answer: Thank you for these suggestions. As you suggested, ROC on PR and ER status were removed from the actual version of the manuscript. Basal phenotype was extracted from the PAM50 classification and multiroc analysis was performed against this outcome on the 11 gene signature. Also, TNBC phenotype was computed from the 3 columns of ER, PR, HER2 status in IHC and TNBC multiroc analyses was performed. Principal component analysis based on eleven gene signature was stratified on these new phenotypes: TNBC and basal.
(4) According to the text, the authors validated the 11-gene ferroptosis/ECM signature in MDA-MB-231 cells treated with erastin and RSL3. However, the dataset used, GSE173905, contains gene expression at 72 hr post treatment of the surviving cells.
Hence, many genes of Fig. 6 cannot be considered as response gene expression but rather may play a compensatory role to prevent ferroptosis and protect the cells from death response. The text must highlight this fact, including sentences from the discussion such as “.. it was observed that RSL3 .. induced overexpression of MET in MDA-MB-231”. The authors can compare their results to datasets such as GSE162069, GSE154425, in which TNBC cells were treated with ferroptosis inducers for shorter time points (therefore showing gene expression in response to the inducers). (5) It is also unclear why the authors choose the GSE173905 dataset to validate the 11 genes, since Figure 6C-E makes it clear that there is no relation between the HR of the 11 genes and their expression in this dataset. The authors didn’t discuss this discrepancy in the discussion.
Answer: The dataset GSE173905 was chosen in order to have a biological validation in term of a transcriptional regulation in a TNBC cellular model which have been stimulated in vitro by two distinct ferroptosis activators. So its meaning that ferroptosis related genes selected by literature and having a worst hazard ratio in breast cancer cohort are effectively regulated by ferroptosis activators, so biologically linked to ferroptosis regulation. As you suggested, supplementary analyses were performed on datasets GSE162069, GSE154425 to test the effect of ferroptosis modifiers at shorter times in vitro and in vivo on the expression of the eleven signature. Thank you for this suggestion. Principal component analyses on these new datasets were also capable to stratified experimental groups.
(6) The 11 genes were chosen based on their positive HR in one dataset. It is very important to validate it in another breast cancer patients datasets, such as METABRIC, which has very good prognostic data (https://www.cbioportal.org/study/summary?id=brca_metabric).
Answer: Thank you for your remark. As you suggested, the eleven gene signature was investigated in METABRIC cohort at multi-omic levels. Patients with alterations on the eleven genes were found to be enriched in ductal and invasive breast so of worst prognosis. Effectively, these affected patients in METABRIC cohort present a worst probability in term of disease free survival. The eleven gene signature stratified METABRIC cohort on several clinical parameters such as pam50 and claudin-low subtype and histological grades. A new figure was integrated in the present manuscript.
Minor comments:
(1) Figure 1A: The authors show a qqplot to display the text mining results. While the distribution of p-values is interesting, other important parameters can be revealed by using another type of plot, for example a scatterplot showing the p-values vs. the number of articles. It might also be beneficial to label the genes with the highest p-values.
Answer: Thank you for your suggestion of this graph which improved the quality of Figure 1 and highlighted the best of genes found by textmining by considering both FDR and number of cited abstracts.
(2) Figure 2C: this analysis is very interesting, however, it might be beneficial to mention the hazard ratios of these 15 genes, to connect these genes to the results shown in 1B.
Answer: A new graph of Hazard ratio for these 15 genes was added showing that are all significant with hazard ratio over one for DRFS outcome.
(3) Figure 4D: Since the 11-gene score was done using the HR values of the genes, the KM plot is unsurprisingly significant. Therefore, it will be interesting to know how this KM, based on the 11 ferroptosis/ECM genes, is compared to a KM plot made using all the 170 genes. Does focusing only on the 11 ECM related genes improves the result over the 170 genes? (4) Figure 4/table 1 – since the authors later work on TNBC models, it will be interesting to show the 11-gene scores in TNBC vs. non-TNBC in Fig. 4 and table 1 (TNBC data can be easily deduced from the clinical data).
Answer: Effectively, TNBC phenotype was computed by aggregating the three columns of IHC for PR, ER and HER2. TNBC status could be calculated for 311 negative patients and 178 positive patients. TNBC principal component analysis based on eleven gene signature of ferroptosis was performed and it shown a good stratification between TNBC positive and negative cases (p-value=1.058352e-27). In table 1, TNBC phenotype was stratified according groups of eleven gene signature and there is a significant increase proportion of TNBC in patients of score HIGH as compared to patients of score LOW (71.8% versus 29.7%, p<1-e4).
(5) The dataset used, GSE25066, contains other transcriptomic based prediction (DLDA-30), as well as residual cancer burden (RCB) prediction. It would be very interesting to compare the authors’ predictions (using the 170 genes or 11 ferroptosis/ECM genes) with these predictions, to increase the impact of the authors signatures.
Answer: Thank you for this suggestion. Eleven gene signature and 170 adverse gene signatures were evaluated against according DLD-30 and RCB therapy response phenotypes. By principal component analysis, 170 gene signature was more robust than eleven one for DLD-30 preoperatory chemotherapy response prediction. Concerning Recurrent cancer burden (RCB) for neoadjuvant therapy only 170 gene signature have discriminant power and especially to stratify RCB0/I classes from RCBII/III classes. The eleven gene signature is not enough weight for RCB prediction.

Reviewer 2 Report
In the manuscript "Adverse Crosstalk Between Extracellular Matrix Remodeling and Ferroptosis in 2 Basal Breast Cancer" the authors should make clear that it is a theoretical study from data mining. Also, the differences between breast cancer patients and control groups should be clarified.
The results and discussion sections should be resumed to be precise and clear.
All the genes related to ferroptosis are only related to ferroptosis? TNF, IL6, SET, CDKN2A, EGFR, HMGB1, KRAS, 25 MET, LCN2, HIF1A, TLR4. This fact should be better explained. The control groups should be included in the abstract.
The Grammar is good.
Author Response
In the manuscript "Adverse Crosstalk Between Extracellular Matrix Remodeling and Ferroptosis in 2 Basal Breast Cancer" the authors should make clear that it is a theoretical study from data mining.
Answer: “By text mining” was explicit at starting session of methods in the abstract of the article
Also, the differences between breast cancer patients and control groups should be clarified.
Answer: Signature of 11 eleven gene Ferroptosis/ECM was studied through MIPANDA server which displayed aggregate RNASEQ data from multi-institutional consortia such as the TCGA, ICGC, GTEX, and CCLE. These supplemental analyses provided in this R1 revision allowed to verified differential expression of 11 genes between normal breast and primary breast cancer but also differential expression between normal breast and metastatic breast cancer. Normal Breast taken as control.
The results and discussion sections should be resumed to be precise and clear.
Answer: Thank you for your suggestion, efforts has done in this sense in actual version.
All the genes related to ferroptosis are only related to ferroptosis? TNF, IL6, SET, CDKN2A, EGFR, HMGB1, KRAS, 25 MET, LCN2, HIF1A, TLR4. This fact should be better explained. The control groups should be included in the abstract.
Answer: All these genes are not strictly related to ferroptosis literature but also connected to the literature related to extracellular remodeling in the context of breast. This is the main interest of the study to focused on adverse expressed genes in breast cancer which are also connected to the microenvironment deregulation in literature. This is the originality of this work to see that genes regulated by ferroptosis activator could also impact on functionalities like extracellular matrix remodeling which could contribute to breast cancer pathogenesis.
For validation of this signature related to ferroptosis in the present version of the manuscript, additional validation showed that of them are already known to belong to FerrDB database version 2. (PMID: 36305834).

Reviewer 3 Report
The onset of breast cancer is increasing worldwide. The triple negative (estrogen receptor, progesterone receptor and HER2) breast cancer is resistant to hormone therapy or chemo, radio-therapy. In this manuscript, Desterke C et al demonstrates in silico analyses. This manuscript is superficial and need to perform additional experiments.
Major points
#1: In figure 1, how did authors define the association between these genes and ferroptosis?
#2: In figure 2C, how did authors select 15 ferroptosis genes as best candidate? There is no p53, SLC7A11, or NF2.
#3: In figure 3B, C, ER or PR are used to define group, while authors discuss triple negative breast cancer.
#4: In figure 6, the data was generated by another group (GSE173905, reference 19). Please examine protein levels of these molecules.
Minor points
##1: In figure 4A and Table 1, what do G1, G2 and G3/4 mean? Are they histological grade?
##2: In figure 4A, B, it is hard to distinguish the group due to color similarity.
Author Response
The onset of breast cancer is increasing worldwide. The triple negative (estrogen receptor, progesterone receptor and HER2) breast cancer is resistant to hormone therapy or chemo, radio-therapy. In this manuscript, Desterke C et al demonstrates in silico analyses. This manuscript is superficial and need to perform additional experiments.
Answer: “By text mining” was explicit at starting session of methods in the abstract of the article.
Major points
#1: In figure 1, how did authors define the association between these genes and ferroptosis?
Answer: A textmining approach was used to query pubmed database by Génie algorithm in first step. This machine learning algorithm allowed prioritized gene citations in pubmed abstracts which have been query by a specific keyword. Here in this context, the keyword “Ferroptosis in breast” was used to subset pubmed corpus and found the gene citations in these selected abstracts: this process was term gene prioritization by textming and this is an intense field used in omics data in last years: https://pubmed.ncbi.nlm.nih.gov/?term=gene+prioritization+and+omics.
In a second step, another text mining algorithm “gene valorization” was used to validated these results and make relations between ferroptosis related genes and ECM remodeling functionality still by querying Pubmed database. Finally, the eleven selected genes were verified as been regulated by RSL3 and erastin ferroptosis activators. Finally, in this revised R1 version, we verified if our 11 selected genes were present in the FerrDB database and that effectively majority of them were present in it (8 of 11).
#2: In figure 2C, how did authors select 15 ferroptosis genes as best candidate? There is no p53, SLC7A11, or NF2.
Answer: These fifteen genes were selected according their largest number of abstracts in pubmed where their corresponding gene symbol are cited based on querying Pubmed database with the following keywords: “ECM-remodeling”, “cancer stem cell (CSC)”, “breast cancer”, “lipid peroxidation”, “regulated cell death”, “ferroptosis”. Eleven genes were retained having highest value of citation both for ferroptosis and ECM-remodeling.
On new Figure 1A, you can see that p53 was found at the top of ferroptosis textmining step 1.
TP53 was found positive for “ferroptosis in breast” text mining at rank 5 with 992 positive abstracts (FDR=0, data not shown).
NF2 was found positive for “ferroptosis in breast” text mining at rank 843 with 12 positive abstracts in pubmed (FDR=2.302e-17, data not shown).
SLC7A11 was found positive for “ferroptosis in breast” text mining at rank 236 with 22 positive abstracts (FDR=1.602e-46, data not shown).
Subsequently, during DRFS survival analysis the expression of these 3 ferroptosis related genes were not found associated to the prognosis of breast cancer patients (Supplemental Table 1).
#3: In figure 3B, C, ER or PR are used to define group, while authors discuss triple negative breast cancer.
Answer: ER and PR multiroc analyses were removed and replaced by basal status.
#4: In figure 6, the data was generated by another group (GSE173905, reference 19). Please examine protein levels of these molecules.
Answer: Thank you for your suggestion, as you asked, we shack the protein level expression of these molecules immune histochemistry on web database of histological tissue for breast cancer on website protein atlas. On TMA of breast cancer tissue samples, we could confirm expression of these markers at protein samples in context of breast cancer.
Minor points
##1: In figure 4A and Table 1, what do G1, G2 and G3/4 mean? Are they histological grade?
Answer: YES. This is the histological tumor grades: G3 and G4 were pooled together in order to have sufficiently samples in this subgroup. This pooling G3 & G4 histological grade respect linear distribution of the cox Residuals during multivariate DRFS survival model and so paste its individual Schoenfeld test.
##2: In figure 4A, B, it is hard to distinguish the group due to color similarity.
Answer: Thank you. We changed the library to perform PCA and used FactoMineR library which implement best color palette and allowed to computed p-values on principal component axes.

Round 2
Reviewer 2 Report
Authors responded all questions and concerns.
Author Response
Once again, thank you very much for your comments and suggestions. Your comments have been very helpful for revising and improving our paper.
Reviewer 3 Report
Authors revised manuscript extensively; however, following point should be clarified.
In figure 6, authors cited images from protein atlas. Is this combined to your in silico analyses? It is hard to understand the relationship.
Author Response
Once again, thank you very much for your comments and suggestions. Your comments have been very helpful for revising and improving our paper.
As you asked previously for #4, to examine protein levels of these molecules, we shack the protein level expression of these molecules immune histochemistry on web database of histological tissue for breast cancer on website protein atlas. On TMA of breast cancer tissue samples, we could confirm expression of these markers at protein samples in context of breast cancer. It is for this reason that we added these images from protein atlas.